# The Effects of Virtual Reality Interventions on Motor Function Rehabilitation in Lower-Limb Amputees: A Systematic Review and Metanalysis

**DOI:** 10.3390/bioengineering12111170

**Published:** 2025-10-28

**Authors:** Jade Paillet, Manuel del Valle Rodríguez, Javier Herranz Vázquez, Francisco Javier Ruiz-Matas Contreras, Julia Raya-Benítez, María Granados Santiago, Marie Carmen Valenza

**Affiliations:** 1Institut de Formation en Masso-Kinésithérapie-Assas, Université Versailles-Saint Quentin en Yvelines, 78280 Paris, France; jade.paillet17@gmail.com; 2Department of Physical Therapy, University of Granada, 18071 Granada, Spain; manueldvrguez@gmail.com (M.d.V.R.); javier7108hv@gmail.com (J.H.V.); fruizmatascontreras@gmail.com (F.J.R.-M.C.); 3Department of Nursing, University of Granada, 18071 Granada, Spain; juliarb@ugr.es (J.R.-B.); mariagranados@ugr.es (M.G.S.)

**Keywords:** lower-limb amputation, virtual reality, motor function, functional mobility

## Abstract

Background: Lower-limb amputation is a complex condition that profoundly affects motor function and patients’ quality of life. Physical therapists are key in managing lower-limb amputees, and emerging technologies such as virtual reality offer promising tools to further enhance motor function. This review aims to assess the effectiveness of virtual reality-based rehabilitation in improving motor function in lower-limb amputees. Method: A systematic review was conducted using the PubMed, Scopus, and Web of Science databases, covering all studies published from their inception to July 2025. The study population consisted of adult lower-limb amputees receiving virtual reality-based rehabilitation, either alone or combined with other interventions. The search strategy included key terms such as “amputee,” “limb loss,” and “virtual reality,” with no date restrictions. Results: Six studies, all randomized controlled trials, were included and featured a variety of protocols. Meta-analysis showed no significant improvement in walking capacity in the experimental group when compared with controls (*p* > 0.05). In contrast, postural stability demonstrated significant improvement in the experimental group, with high consistency across studies (I^2^ = 0%). Conclusions: Current evidence suggests that virtual reality may be an effective approach to improving motor function in lower-limb amputees. However, the studies exhibit methodological limitations, highlighting the need for further research to standardize protocols and evaluate long-term benefits.

## 1. Introduction

Lower extremity amputation remains a significant healthcare issue. Estimates from the largest inpatient care database suggest that approximately 115,000 individuals undergo this procedure annually in the United States, and 50,000 to 60,000 of these individuals undergo major amputations, with such procedures defined as extending above the ankle [1]. Major amputations are most frequently performed to remove necrotic tissue, with common causes including diabetes mellitus, peripheral arterial disease, bone and joint infections, peripheral neuropathy, trauma, and malignancy [2]. The rising prevalence of obesity and vascular disorders has led to projections that by 2050, the prevalence of limb loss will more than double, affecting an estimated 3.6 million individuals [3].

Irrespective of etiology, lower-limb amputation often results in significant functional impairment, largely due to a combination of personal factors and physical deficits such as altered balance, reduced strength, and compromised mobility. Early, active rehabilitation, including both physical and occupational therapy, is crucial for optimizing recovery [4]. These interventions encourage prosthesis use and facilitate the resumption of routine social activities, all of which are essential for restoring independence and quality of life. Notably, studies such as those by Dillingham et al. have demonstrated that the type and intensity of rehabilitation setting can influence functional outcomes for amputees [5,6].

Daily functioning can be profoundly affected in individuals with lower-limb amputation, often leading to reduced satisfaction and overall well-being. In response, a growing body of research has focused on rehabilitation strategies aimed at restoring physical abilities through structured, repetitive movement programs. Traditional rehabilitation commonly involves equipment, such as treadmills or weights, to provide resistance during movement exercises. Targeted outcomes for individuals with lower-limb impairment include improvements in walking capacity, either independently or with assistive devices, and enhanced postural stability during static and dynamic activities [7,8].

More recently, technological advances have introduced devices capable of delivering immersive visual feedback synchronized with physical movement, with virtual reality (VR) emerging as the most promising modality [9].

However, systematic reviews highlight that the effectiveness of VR training as a component of rehabilitation has not been widely explored in the lower-limb amputee population. Most studies are limited by small sample sizes, heterogeneous methodologies, and insufficient statistical power; these factors collectively reduce the reliability of the evidence [10,11]. Despite these limitations, VR-based therapeutic exercise presents opportunities for more engaging, efficient, and versatile movement practice in settings that would otherwise be logistically challenging or physically demanding to replicate.

To our knowledge, no systematic review has synthesized the literature on therapeutic exercise interventions through virtual reality in lower-limb amputation patients. Therefore, the purpose of this systematic review and meta-analysis was to summarize the use of VR as part of therapeutic exercise interventions on motor function in individuals with lower-limb amputation to provide an up-to-date overview of intervention studies. The specific objectives were to (1) describe the characteristics of these interventions and (2) assess their effects on motor function in patients with lower-limb amputations.

## 2. Materials and Methods

The Preferred Reporting Items for Systematic Reviews and Meta-Analyses (PRISMA) [12] statement guidelines were followed when conducting this systematic review and meta-analysis (Appendix A). The Cochrane Collaboration guidelines were used as a reference, and we also registered our protocol in the PROSPERO database with the code CRD42025572713.

### 2.1. Search Strategy

The search strategy used was developed for MEDLINE, and considered: (1) keywords and key terms used in other existing systematic reviews on the same topic; (2) the use of MeSH terms following a thorough examination of the database; and (3) consultation with a specialist to guide and review the search strategy. Our search equation was adapted to PubMed, Web of Sciences, and Scopus databases. The search included studies published from the inception of each database up to July 2025, with the literature search conducted between June and July 2025. Finally, the reference lists of previous reviews were considered, and the search strategy was refined in the various databases mentioned by means of search tests. A comprehensive description of the search strategy is provided in Appendix A.

### 2.2. Inclusion and Exclusion Criteria

Based on the participants, interventions, outcomes, and study design, a research question was formulated based on the PICOS model [13]. Based on this model, the applied inclusion criteria were as follows: (1) Adults patients with lower-limb amputation. (2) Therapeutic exercise provided through any virtual reality modality. (3) Control intervention not including therapeutic exercise through virtual reality or no-treatment intervention group. (4) As an outcome study variable, lower-limb motor function was defined as the ability to produce voluntary body movements through the coordinated activity of the brain, motor neurons, and muscles. Ambulation refers to the capacity to walk, either independently or with assistive devices, while balance describes the ability to maintain postural stability and prevent falls during static and dynamic physical activities [8]. (5) Randomized clinical trials.

The selection of virtual reality interventions was made according to the definition established by Schultheis and Rizzo [14] as “an advanced form of human-computer interface that allows the user to interact naturally with a computer-assisted environment.” Additionally, besides virtual reality glasses, virtual reality was also considered when delivered though wall projection or interactive gamification. Different degrees of immersion were taken into account: Moderate immersion is achieved using a wide curved screen or by projecting screens onto walls [15,16]. In low-immersion setups, technology is typically limited to a traditional monitor or television screen, resulting in a significantly weaker sensory connection compared with high-immersion VR [17].

### 2.3. Study Selection and Literature Data Extraction

After the process of obtaining records from the different databases was completed, the process of eliminating duplicate studies was carried out. Then, three reviewers (JHV, FJRMC, and MVR) independently performed the selection of studies based on the titles and abstracts of the studies to ensure their eligibility. In case of discrepancies, a fourth reviewer resolved the inclusion doubts.

The literature screening was conducted by three researchers and based on predetermined inclusion and exclusion criteria. Data extraction was performed independently using a pre-designed standardized data extraction sheet in Microsoft Excel. Extracted data included basic details of the included trials, such as the lead author and year of publication; basic characteristics of the study participants, including the number of patients, age distribution, and gender composition; specifics of the intervention, such as virtual reality (VR) equipment used, level of immersion stem, and duration of the intervention; and results of the main motor function assessments.

### 2.4. Literature Quality Assessment

After the selection of studies, data extraction and methodological assessment of the studies were performed independently by two reviewers and a third reviewer who resolved discrepancies. The methodological quality of the studies was assessed using the PEDro scale [18], which consists of 11 criteria. The first criterion relates to the external validity of the study and is not included in the total score calculation. Criteria 2 to 9 evaluate the internal validity of the study. Criteria 10 and 11 determine whether the studies contain statistical information allowing for a correct interpretation of the results. A high PEDro score does not measure the validity of a study’s conclusions but indicates methodological quality and the likelihood that the results are reliable. Scores were interpreted as follows: Excellent (9–10), Good (6–8), Fair (4–5), and Poor (≤3).

### 2.5. Risk of Bias Analysis

The Cochrane risk of bias tool was used to assess the risk of bias for each of the clinical trials included in this review. Following its guidelines, we assessed the risk of bias of the subscale’s selection bias performance, detection bias, attrition bias, reporting bias, and other biases. Based on the risk of bias of each subscale, we obtained a final score that allowed us to classify the included studies into low risk of bias (if all subscales were at a low risk of bias), some concern of bias (if one or two subscales were unclear), or high risk of bias (if one scale had limitations that invalidated the results or if more than two subscales were unclear) [19].

### 2.6. Meta-Analysis

Quantitative analysis was conducted using the Review Manager 5 (Rev-Man version 5.1, updated March 2011) software for all studies presenting post-intervention means and standard deviations. The mean difference value was used to assess postural stability and walking capacity. Data, including final mean values, standard deviations, and the number of patients assessed at different endpoints for each treatment arm, were extracted to estimate overall mean differences between treatment arms.

For articles with insufficient data to calculate effect size (e.g., no provided means or standard deviation), authors were contacted to obtain the required information. When *p*-values or 95% confidence intervals were available, and standard deviations were missing, calculations were performed following the guidelines outlined in the Review Manager manual [20]. These measures were implemented to maximize the reliability and validity of the findings.

Continuous outcomes were analyzed using weighted mean differences when all studies measured outcomes on the same scale. Standardized mean differences were employed when different scales were assumed to measure the same underlying condition. The 95% confidence intervals were computed for all outcomes. Overall mean effect sizes were estimated using random effect models or fixed effect models based on I^2^ tests for statistical heterogeneity. I^2^ < 50% is considered to be a meta-analysis with low heterogeneity, and a fixed-effects model was used [21]. A visual inspection of the forest plots for outlier studies was also performed.

## 3. Results

### 3.1. Study Selection

A total of 5260 records were initially identified through database searching. The screening of titles and abstracts resulted in 19 full-text articles for comprehensive review. No additional studies were found through alternative methods. A total of six studies were included in the qualitative and quantitative syntheses [22,23,24,25,26,27]. The PRISMA flow chart is illustrated in Figure 1.

### 3.2. Study Characteristics

Of the six studies included, four were rated as being of good quality according to the PEDro scale [22,23,24,26], while two were considered to be of fair quality. As shown in Figure 2, the RoB 2 assessment revealed that three studies had some concerns regarding the risk of bias [22,23,24], and these were classified as being high risk [25,26,27]. Notably, the predominance of small sample sizes and the lack of blinding across most studies markedly reduces the external validity of the current evidence base. High-risk studies, especially those with methodological flaws such as inadequate blinding or low participant numbers, have the potential to overestimate treatment effects and compromise the reliability and generalizability of the findings. Therefore, these methodological limitations should be considered when interpreting the overall impact of VR interventions in this population.

Table 1 shows the main characteristics of the included amputation patients. A total of 199 amputation patients were included in this systematic review. Many patients were male (76.38%), with a mean age of 48.76 years old. There were three types of amputations in the included studies: unilateral transtibial amputation [22,23,24,25,26,27], unilateral transfemoral amputation [22,23,24,27], or knee disarticulation [22,27].

Details about virtual reality-based rehabilitation treatments and obtained results are reported in Table 2.

There were virtual reality-based rehabilitation treatments based on mixed modalities [22,23,25,26], and biomechanical exercises [24,27]. The duration of the interventions ranged between four weeks [22,25,26,27], six weeks [23], and eight weeks [24]. The VR level of immersion was primarily based on exergames using Kinect^TM^ [23,25,26] or Wii^TM^ [22,24], while one study employed immersive VR [27]. Lower-limb motor function was evaluated with different outcomes: Two Minute Walk Test, SPPB (2MWT) [22,24,26]; Short Physical Performance Battery (SPPB) [22,24]; Physical Activity Scale for the Elderly (PASE) [22]; Activities-specific Balance Confidence (ABC) [22,24,27]; Modus Health Stepwatch^TM^ Activity Monitor (SAM) [22]; Walking While Talking Test (WWT) [22]; Locomotor Capabilities Index in Amputees (LCI-5) [22]; 6 min walk test (6 MWT) [23,25]; Berg Balance Scale (BBS) [23]; Dynamic Gait Index (DGI) [23]; Timed Up and Go test (TUG) [23,26]; Four-Step Square Test (FSST) [24]; 10-Meter Walk Test (10MWT) [25]; amputee mobility predictor with prosthesis (AMPRO) [26]; physiological cost index (PCI) [26]; cadence [25]; and single leg balance test [25].

An improvement in lower-limb motor function was observed in four studies, which reported significant differences between groups, and the results were in favor of the intervention groups (*p* < 0.05) [23,24,25,26]; however, two studies [22,27] did not report significant differences.

### 3.3. Results of Meta-Analysis

The results for walking capacity have been analyzed as shown in Figure 3. The pooled mean difference (MD) showed no significant overall effect when comparing the experimental group with the control group [MD = 0.13, 95% confidence interval (CI): −0.47, 0.72; *p* = 0.68]. The meta-analysis showed a moderate-to-high level of heterogeneity (I^2^ = 69%), indicating variability among the included studies.

The results for postural stability have been analyzed as shown in Figure 4. The pooled MD showed a significant overall effect when the experimental group was compared with the control group [MD = 8.03, 95% confidence interval (CI): 0.98, 15.07; *p* = 0.03]. There was no evidence of heterogeneity among the included studies (I^2^ = 0%), indicating a high level of consistency in the effect estimates.

## 4. Discussion

The aim of this meta-analysis was to determine the effects of virtual reality-based rehabilitation treatments in lower-limb amputation patients. We found that virtual reality-based rehabilitation treatments are effective in enhancing lower-limb motor function, demonstrating improvements in postural stability.

The most common type of amputation was unilateral transtibial amputation. Epidemiological studies consistently show that transtibial amputations account for approximately 50–60% of all major lower-limb amputations [28]. This predominance is largely attributed to the epidemiology of underlying etiologies, primarily diabetes mellitus and peripheral arterial disease, which often result in distal limb ischemia or infection while sparing sufficient viable tissue below the knee. When feasible, a transtibial level is preferred over more proximal amputations because preservation of the knee joint offers significant functional advantages [29,30].

Our results observed a significant improvement in postural stability in studies that used the Activities-specific Balance Confidence (ABC) Scale to measure primary outcomes. The meta-analysis of postural stability showed a significant pooled mean difference of 8.03 points favoring the experimental interventions, with no heterogeneity. This finding suggests that VR-based interventions improved postural stability when compared to the control groups. However, the confidence interval is close to zero, and previous studies have reported minimal clinically important differences ranging from 8.64 to 16.94 points on the ABC Scale in populations with vestibular disorder [31]. Therefore, this improvement should be interpreted as modest in clinical terms.

VR enables enriched sensory feedback, high-repetition gait cycles, immersive and motivating environments, and the opportunity to manipulate parameters such as optic flow, step length or obstacle negotiation, all of which are known to drive gait improvements in populations with stroke survivors [32,33,34]. However, our findings did not indicate an improvement in walking capacity following the VR-based intervention in individuals with lower-limb amputation. A systematic review of VR interventions for people with amputations [10] found that while VR can improve balance and some gait-related outcomes, most included studies used non-immersive or semi-immersive methods, featuring simple games or environments that do not fully replicate the biomechanical and sensory demands of walking. Additionally, most studies included in our study implemented mixed-modality or biomechanical VR interventions. Therefore, the absence of aerobic activities in VR protocols is a plausible explanation for these results. In our results, the presence of aerobic exercises in the control group of Sahan’s study [25] may account for the observed improvement in walking capacity compared to the experimental VR intervention. Improvements in walking capacity rely heavily on cardiovascular conditioning and muscular endurance, which require sustained, repetitive, and progressive locomotor activity [35]. Without tasks that challenge aerobic thresholds, patients are unlikely to experience the cardiovascular and metabolic adaptations necessary to extend walking distance or duration [36,37].

To optimize the impact of VR interventions on walking capacity, future protocols should explicitly incorporate endurance or aerobic components, such as virtual scenarios that require prolonged walking, progressive step counts, and task-oriented cardiovascular challenges [38]. These features can help stimulate the physiological and metabolic demands essential for walking endurance, which are currently missing from many VR-based rehabilitation platforms. Notably, long-term and high-frequency VR-based therapies, particularly those exceeding 20 sessions, have been shown to yield greater gains in functional mobility and endurance in lower-extremity rehabilitation populations [39].

Furthermore, current VR programs tend to emphasize postural control, proprioceptive feedback, and functional engagement over endurance-based locomotion. While these domains are important for overall mobility, they may not directly transfer to improvements in community ambulation, especially when measured through capacity tests like the 6MWT [10,40,41,42]. This aligns with the conclusions of our review, which found that VR interventions can be as effective as conventional rehabilitation for improving balance in individuals with lower-limb amputation, but the evidence for enhancing walking capacity remains inconclusive.

Although VR has shown promise as an adjunct to conventional rehabilitation in lower-limb amputees, several limitations must be addressed before it can be widely implemented in clinical practice. Current systems primarily emphasize visual and auditory immersion but provide insufficient feedback to support proprioceptive and motor retraining, both of which are critical for effective prosthetic use. Consequently, functional recovery may be constrained. Furthermore, immersive systems remain costly and technically demanding [43]. A study found that initially, VR is costly, owing to the upfront investment in equipment and software development; however, with repeated use, these fixed costs are amortized, making VR a more economical option in the long term [44]. In addition, VR can be a cost-saving adjunct to neurological rehabilitation by reducing professional burden and transportation costs, especially when delivered remotely [45]. Non-immersive systems are consistently cheaper, more accessible, and show clinical improvements comparable to or better than immersive/high-cost systems for many outcomes in upper-limb motor function in stroke survivors [46]. This may explain why the studies included in our review mostly used exergames with non-immersive systems such as Wii Fit and Kinect. Future research should conduct cost-effectiveness analyses comparing low-cost/non-immersive systems, high-cost immersive systems, and conventional therapy in this population, to observe real-world implementation in healthcare.

Importantly, the potential of VR-based rehabilitation is greatly expanded by telerehabilitation platforms, which facilitate home-based, accessible training opportunities and ongoing patient engagement. However, adherence to these at-home programs is a significant concern, as participation typically declines in the absence of structured professional supervision. To overcome this limitation, strategies such as regular remote supervision by physiotherapists, incorporation of behavioral support (e.g., motivational feedback, goal setting, automated reminders), and progress tracking should be embedded within telerehabilitation programs to sustain long-term participation and optimize outcomes [47,48].

Our results are broadly consistent with the latest systematic reviews. Arazpour et al., 2025 [49] found that non-immersive virtual reality/video-game–based training can improve balance-related clinical test results in lower-limb amputees, with semi-immersive approaches showing promise for gait symmetry. Similarly, Hao et al., 2023 [10] concluded that virtual reality improves motor outcomes in amputation rehabilitation, but that evidence is insufficient due to few adequately powered randomized clinical trials and mixed results.

Virtual reality should be considered a complementary tool within multidisciplinary rehabilitation programs, including traditional in-person therapy and telerehabilitation that can accelerate recovery in the short term. Programs may prioritize non-immersive or semi-immersive platforms and consider immersive protocols, considering patient tolerance and preferences. Importantly, telerehabilitation and exergame programs can help sustain training at home, but adherence often declines without professional support, highlighting the need for regular supervision or behavioral strategies [50]. Additionally, VR interventions are difficult to blind; therefore, blinding of participants and personnel was assessed as having a high risk of bias in all studies, representing an inherent limitation of this technology [51]. Future trials should prioritize the blinding of outcome assessors to reduce bias and strengthen the methodological validity of the findings.

Several limitations of this systematic review and meta-analysis are evident. Firstly, the quality of the studies included was not entirely optimal due to the lack of blinding among the professionals, which could influence the analysis, and future randomized controlled trials should adopt more robust study designs to further minimize the risk of bias. Secondly, there was significant heterogeneity in the walking capacity analysis, indicating that the variability among studies may limit the generalizability of our conclusions. Additionally, we were unable to analyze all types of interventions in subgroups in the meta-analyses due to increased heterogeneity. It is also worth noting that the age range of participants across studies varied considerably, which may have subtle implications given that factors such as neuroplasticity and adaptability to technology can differ with age. The exclusion of other health databases may have limited the scope of the review, although many studies retrieved from these sources did not meet the inclusion criteria. Finally, this meta-analysis only examined short-term effects due to a lack of follow-up data in the included studies.

## 5. Conclusions

The effects of VR-based rehabilitation in patients with lower-limb amputation indicate that such interventions are effective in enhancing lower-limb motor function and are associated with improvements in postural stability. However, this conclusion is based mainly on preliminary data, highlighting the need for further research to confirm these effects in individuals with lower-limb amputation. Future research should prioritize the development and adoption of standardized VR rehabilitation protocols to enhance comparability across studies and establish best practice guidelines. It is also recommended that larger, multicenter randomized controlled trials with long-term follow-up be conducted to better determine the lasting effects and real-world applicability of VR interventions. Additionally, integrating immersive feedback modalities such as proprioceptive and haptic input, alongside visual and auditory immersion, may further enhance the therapeutic experience and outcomes, offering a more comprehensive and engaging rehabilitation environment for patients.

## Figures and Tables

**Figure 1 bioengineering-12-01170-f001:**
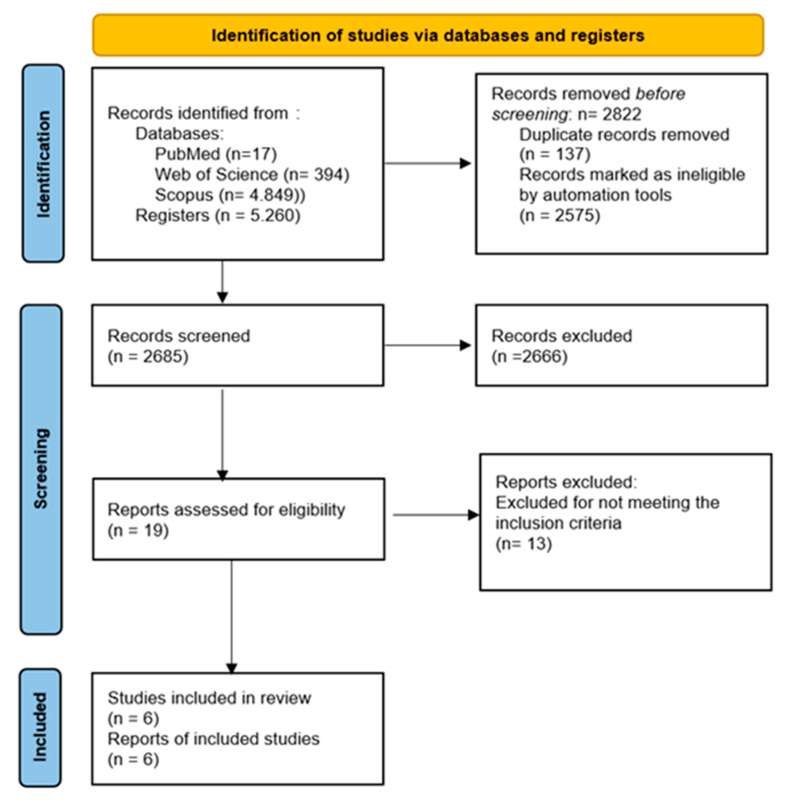
PRISMA flow chart.

**Figure 2 bioengineering-12-01170-f002:**
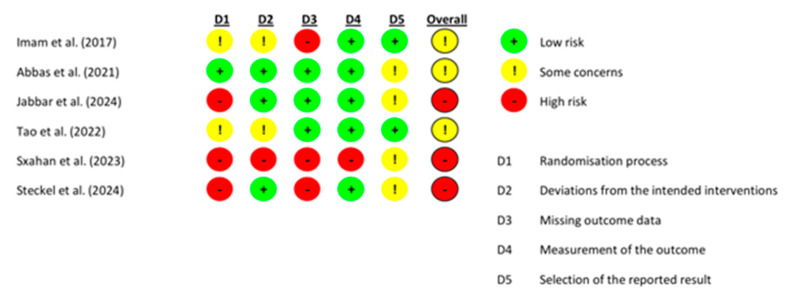
Risk of bias assessment of the included studies [22,23,24,25,26,27].

**Figure 3 bioengineering-12-01170-f003:**
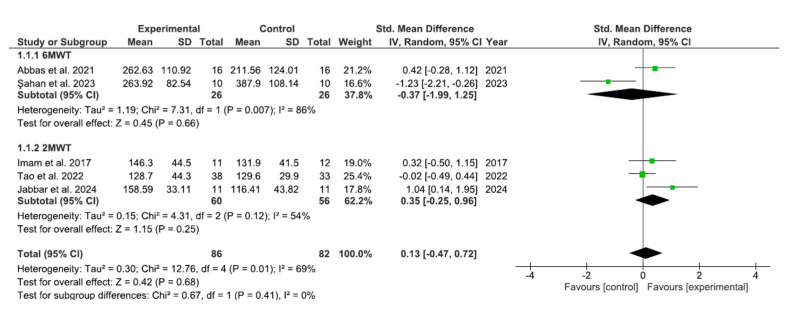
Meta-analysis: forest plot illustrating changes in walking capacity [22,23,24,25,26].

**Figure 4 bioengineering-12-01170-f004:**
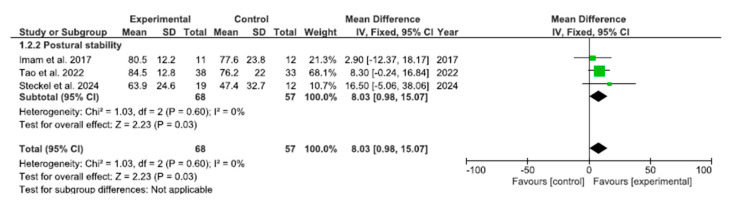
Meta-analysis: forest plot illustrating changes in postural stability [22,24,27].

**Table 1 bioengineering-12-01170-t001:** Characteristics of patients with amputations included in the studies.

Study	Sample Size	Type of Amputation	Sample Age Mean (Male/Female Sex)	PEDro Scale
Imam et al., 2017 [22]	IG:11 CG: 12	Unilateral transtibial amputation, unilateral transfemoral amputation, or knee disarticulation.	IG: 61.5 (12/2) CG: 62.5 (6/8)	8/10
Abbas et al., 2021 [23]	IG: 16 CG: 16	Unilateral transtibial amputation or unilateral transfemoral amputation.	IG: 27.62 (15/1) CG: 27.62 (14/2)	7/10
Tao et al., 2022 [24]	IG: 38 CG: 33	Unilateral transtibial amputation or unilateral transfemoral amputation.	IG: 66.6 (31/7) CG: 63.2 (30/3)	7/10
Şahan et al., 2023 [25]	IG: 10 CG: 10	Unilateral transtibial amputation.	IG: 34 (10/0) CG: 32 (10/0)	4/10
Jabbar et al., 2024 [26]	IG: 11 CG: 11	Unilateral transtibial amputation.	IG: 48.91 (NR) CG: 48.73 (NR)	7/10
Steckel et al., 2024 [27]	IG: 19 CG: 12	Unilateral transtibial amputation, unilateral transfemoral amputation, or knee disarticulation.	IG: 52.7 (16/3) CG: 59.8 (8/4)	5/10

IG: Intervention group; CG: Control group; NR: not reported.

**Table 2 bioengineering-12-01170-t002:** Characteristics of virtual reality-based rehabilitation treatments in patients with amputations.

Study	Intervention Group	Control Group	Timing	Classification of Exercise Interventions	VR Level of Immersion	Outcomes of Lower-Limb Motor Function
Imam et al., 2017 [22]	Wii Fit training, balance, yoga, strength, and aerobics games	Big Brain Academy cognitive video games	IG: 3 times a week for 4 weeks CG: 3 times a week for 4 weeks	IG: Mixed modality CG: Cognitive	Exergame/Wii^TM^	2MWT: IG > CG (*p* > 0.05); ABC: IG > CG (*p* > 0.05); SPPB: IG > CG (*p* > 0.05); SAM: IG > CG (*p* > 0.05); PASE: IG > CG (*p* > 0.05); LCI-5: IG < CG (*p* > 0.05); WWT-simple: IG > CG (*p* > 0.05); WWT-complex: IG > CG (*p* > 0.05)
Abbas et al., 2021 [23]	VR balance and gait training + conventional rehabilitation	Conventional rehabilitation only	IG: 3 times a week for 6 weeks CG: 3 times a week for 6 weeks	IG: Mixed modality CG: Biomechanical	Exergame/Kinect^TM^	BBS: IG > CG (*p* < 0.05); TUG: IG > CG (*p* < 0.001); DGI: IG > CG (*p* < 0.001); 6MWT: IG > CG (*p* > 0.05)
Tao et al., 2022 [24]	WiiNWalk telerehabilitation	Big Brain Academy cognitive video games	IG: 4 weeks supervised + 4 weeks unsupervised CG: NR	IG: Biomechanical CG: Cognitive	Exergame/Wii^TM^	2MWT: IG > CG (*p* < 0.05); SPPB: IG > CG (*p* > 0.05); FSST: IG > CG (*p* < 0.05); ABC: IG > CG (*p* < 0.05)
Şahan et al., 2023 [25]	Interactive VR exergames	Conventional prosthetic rehabilitation	IG: 3 times a week, 45 min sessions, 4 weeks CG: NR	IG: Mixed modality CG: Biomechanical and aerobic	Exergame/Kinect^TM^	6MWT: IG < CG (*p* < 0.05); single leg balance test (prosthesis limb): IG < CG (*p* < 0.05); single leg balance test (healthy limb): IG > CG (*p* > 0.05); 10MWT: IG > CG (*p* < 0.05); Cadence: IG > CG (*p* < 0.05)
Jabbar et al., 2024 [26]	Exergames + conventional rehabilitation	Conventional rehabilitation	IG: 3 times a week for 4 weeks CG: 3 times a week for 4 weeks	IG: Mixed modality CG: Biomechanical	Exergame/Kinect^TM^	2MWT: IG > CG (*p* > 0.05); TUG: IG > CG (*p* < 0.05); AMPRO: IG > CG (*p* > 0.05); PCI: IG < CG (*p* > 0.05).
Steckel et al., 2024 [27]	Immersive VR + standard rehabilitation	Standard rehabilitation	IG: 3 times a week, 30 min sessions for 4 weeks CG:NR	IG: Biomechanical CG: NR	Immersive/High	ABC: IG < CG (*p* > 0.05)

IG: Intervention group; CG: Control group; NR: Not reported; VR: Virtual reality; 2MWT: Two Minute Walk Test; SPPB: Short Physical Performance Battery; PASE: Physical Activity Scale for the Elderly; ABC: Activities-specific Balance Confidence; SAM: Modus Health Stepwatch^TM^ Activity Monitor; WWT: Walking While Talking Test; LCI-5: Locomotor Capabilities Index in Amputees; 6 MWT: 6 min walk test; BBS: Berg Balance Scale; DGI: Dynamic Gait Index; TUG: Timed Up and Go test; FSST: Four-Step Square Test; 10MWT: 10-Meter Walk Test; AMPRO: Amputee mobility predictor with prosthesis; PCI: Physiological cost index.

## Data Availability

No new data were created or analyzed in this study.

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
