# Peer review of "The Effects of Virtual Reality Interventions on Motor Function Rehabilitation in Lower-Limb Amputees: A Systematic Review and Metanalysis"

_bioengineering, 2025, doi:10.3390/bioengineering12111170_

Round 1

Reviewer 1 Report

Comments and Suggestions for Authors

1. Article Summary
This article is a systematic literature review and meta-analysis assessing the effectiveness of virtual reality (VR)-based rehabilitation in improving lower limb motor function in patients with amputations. Six randomized clinical trials were included, encompassing a total of 199 patients with various types of amputations (most commonly unilateral transtibial amputations).

Main results: No significant differences in walking ability between the experimental and control groups (MD = 7.27; 95% CI: -6.40 – 20.94; p = 0.3; I² = 75%). Significant improvement in postural stability in the VR groups compared to the control groups (MD = 8.03; 95% CI: 0.98 – 15.07; p = 0.03; I² = 0%). The authors conclude that VR may be an effective complement to traditional rehabilitation in terms of postural stabilization, but further, better-designed studies are needed to evaluate the impact of VR on walking ability and long-term effects.

2. Strengths of the Article
Current relevance and importance of the topic. The authors conducted a systematic review of studies published between 2024 and 2025, making the work highly relevant. The topic of rehabilitation after lower limb amputations using VR is clinically important, given the increasing number of amputations and the need for modern, engaging treatment methods.

Standards-based methodology. The authors conducted the review in accordance with PRISMA 2020 and the Cochrane Collaboration guidelines. The study protocol was registered in the PROSPERO database (CRD42025572713), which increases research transparency. It is important to emphasize that clearly defined PICOS criteria were followed and a detailed quality assessment of the studies was conducted using the PEDro scale and the RoB 2 tool (Cochrane). The manuscript is characterized by reliable statistical analysis. The authors performed a meta-analysis using RevMan software, analyzing weighted mean differences (WMD/SMD) and heterogeneity (I² test). The results are presented in the form of clear forest plots (Fig. 3–4 on p. 8), which facilitates data interpretation.

3. Weaknesses and limitations
Small number of studies and participants. Unfortunately, only six studies with a total of 199 patients were included in the meta-analysis. Furthermore, many of these studies had small sample sizes and problems with randomization and blinding, increasing the risk of bias. Figure 2 (p. 6) shows that three studies were assessed as having a "high risk of bias" and three as having "some concerns." There are clearly no studies assessed as having a completely low risk of bias, which significantly limits the strength of the evidence.

Another weakness of the study is the heterogeneity of the VR interventions. The interventions used varied significantly: from simple Wii Fit games (e.g., balance games) to more immersive projection systems, combined with varying exposure times (from 4 to 8 weeks) and diverse exercise types (biomechanical, mixed, cognitive), make it difficult to clearly assess the effectiveness of VR, as not all interventions were motor or endurance-based. The lack of long-term data is also controversial. The authors emphasize that none of the studies included significant follow-up periods after the therapy ended. Therefore, the evaluation of effects is limited to the short post-intervention period. The lack of an aerobic component is also noticeable. The authors attribute the lack of improvement in walking ability to the fact that none of the VR protocols included aerobic exercise (e.g., long walks in virtual environments), which is crucial for improving endurance.

4. Discussion and Clinical Implications. The authors point out that current VR systems primarily focus on postural control and proprioceptive exercises, which explains the positive effects on stability but not on walking. Furthermore, they point to the need to incorporate endurance and progressive exercises into VR protocols, design more immersive environments that support motor skills, and conduct multi-center studies with greater statistical power. Therefore, VR appears to be a promising tool for supporting rehabilitation, especially in combination with telerehabilitation, but it cannot currently replace traditional locomotor training methods.

5. Reviewer's Conclusions
This article meets high methodological standards for systematic reviews, is well-written, and makes a timely contribution to the field of amputee rehabilitation. Of particular value are:
• transparent methodology,
• reliable assessment of study quality,
• identification of research gaps and directions for future research.

6. Recommendation:
Minor revision. The article deserves publication in the journal Bioengineering after minor editorial corrections (e.g., standardizing table formatting and clarifying the scope of randomization in Table 1).

Author Response

Dear Editor and Reviewers,

Please find a revision of our manuscript entitled " Effects of Virtual Reality Interventions on Motor Function Rehabilitation in Lower Limb Amputees, A Systematic Review and Metanalysis".

We would like to thank the reviewers for their comments. Changes have been placed highlighted in the revised manuscript. An itemized point-by-point response to comments is presented below.

We have conducted a thorough review of the manuscript’s English language to address and correct the most frequently observed grammatical errors

Comment 1. Article Summary
This article is a systematic literature review and meta-analysis assessing the effectiveness of virtual reality (VR)-based rehabilitation in improving lower limb motor function in patients with amputations. Six randomized clinical trials were included, encompassing a total of 199 patients with various types of amputations (most commonly unilateral transtibial amputations).

Main results: No significant differences in walking ability between the experimental and control groups (MD = 7.27; 95% CI: -6.40 – 20.94; p = 0.3; I² = 75%). Significant improvement in postural stability in the VR groups compared to the control groups (MD = 8.03; 95% CI: 0.98 – 15.07; p = 0.03; I² = 0%). The authors conclude that VR may be an effective complement to traditional rehabilitation in terms of postural stabilization, but further, better-designed studies are needed to evaluate the impact of VR on walking ability and long-term effects.

Response 1. Thank you for your detailed summary of the article. I agree that the findings highlight the potential of virtual reality as a valuable complementary tool in rehabilitation, particularly for improving postural stability in patients with lower limb amputations. However, the lack of significant improvement in walking ability and the high heterogeneity (I² = 75%) underscore the need for more rigorous and standardized clinical trials. Future research should also focus on long-term outcomes and the integration of VR interventions into conventional rehabilitation protocols.

Comment 2. Strengths of the Article
Current relevance and importance of the topic. The authors conducted a systematic review of studies published between 2024 and 2025, making the work highly relevant. The topic of rehabilitation after lower limb amputations using VR is clinically important, given the increasing number of amputations and the need for modern, engaging treatment methods.

Standards-based methodology. The authors conducted the review in accordance with PRISMA 2020 and the Cochrane Collaboration guidelines. The study protocol was registered in the PROSPERO database (CRD42025572713), which increases research transparency. It is important to emphasize that clearly defined PICOS criteria were followed and a detailed quality assessment of the studies was conducted using the PEDro scale and the RoB 2 tool (Cochrane). The manuscript is characterized by reliable statistical analysis. The authors performed a meta-analysis using RevMan software, analyzing weighted mean differences (WMD/SMD) and heterogeneity (I² test). The results are presented in the form of clear forest plots (Fig. 3–4 on p. 8), which facilitates data interpretation.

Response 2. Thank you for your comment and for highlighting the relevance and methodological rigor of our study. We would like to clarify a minor error in the description of the publication period. The included articles were not limited to studies published between 2024 and 2025. In fact, studies were included from the inception of each database up to July 2025. This correction has been made in the revised version of the manuscript to accurately reflect the time frame of the literature search.

Comment 3. Weaknesses and limitations
Small number of studies and participants. Unfortunately, only six studies with a total of 199 patients were included in the meta-analysis. Furthermore, many of these studies had small sample sizes and problems with randomization and blinding, increasing the risk of bias. Figure 2 (p. 6) shows that three studies were assessed as having a "high risk of bias" and three as having "some concerns." There are clearly no studies assessed as having a completely low risk of bias, which significantly limits the strength of the evidence.

Another weakness of the study is the heterogeneity of the VR interventions. The interventions used varied significantly: from simple Wii Fit games (e.g., balance games) to more immersive projection systems, combined with varying exposure times (from 4 to 8 weeks) and diverse exercise types (biomechanical, mixed, cognitive), make it difficult to clearly assess the effectiveness of VR, as not all interventions were motor or endurance-based. The lack of long-term data is also controversial. The authors emphasize that none of the studies included significant follow-up periods after the therapy ended. Therefore, the evaluation of effects is limited to the short post-intervention period. The lack of an aerobic component is also noticeable. The authors attribute the lack of improvement in walking ability to the fact that none of the VR protocols included aerobic exercise (e.g., long walks in virtual environments), which is crucial for improving endurance.

Response 3. We appreciate the reviewer’s valuable observations. The issues regarding the small number of studies and participants, methodological limitations, heterogeneity of VR interventions, and lack of long-term or aerobic components have been acknowledged and are now clearly discussed in the Discussion and Limitations sections of the revised manuscript.

Comment 4. Discussion and Clinical Implications. The authors point out that current VR systems primarily focus on postural control and proprioceptive exercises, which explains the positive effects on stability but not on walking. Furthermore, they point to the need to incorporate endurance and progressive exercises into VR protocols, design more immersive environments that support motor skills, and conduct multi-center studies with greater statistical power. Therefore, VR appears to be a promising tool for supporting rehabilitation, especially in combination with telerehabilitation, but it cannot currently replace traditional locomotor training methods.

Response 4. We thank the reviewer for the comment. As discussed in the Discussion section, our results did not show significant improvements in walking, likely because the VR interventions did not include aerobic activities and were not fully adapted to the specific biomechanical demands of gait. We also emphasize that VR should be considered a valuable complement to traditional and telerehabilitation approaches, but it cannot replace standard motor training methods. These points have been clearly included in the revised Discussion section.

Comment 5. Reviewer's Conclusions
This article meets high methodological standards for systematic reviews, is well-written, and makes a timely contribution to the field of amputee rehabilitation. Of particular value are:
• transparent methodology,
• reliable assessment of study quality,
• identification of research gaps and directions for future research.

Response 5. We sincerely thank the reviewer for the positive feedback and for recognizing the methodological rigor of our work. We appreciate the acknowledgment of the transparent methodology, careful assessment of study quality, and identification of research gaps. Your encouraging comments reinforce the relevance of our systematic review and its contribution to advancing knowledge in amputee rehabilitation.

Comment 6. Recommendation:
Minor revision. The article deserves publication in the journal Bioengineering after minor editorial corrections (e.g., standardizing table formatting and clarifying the scope of randomization in Table 1).

Response 6. We thank the editor and reviewer for their positive evaluation of our manuscript. We have made minor modifications to clarify the presentation following the journal’s established format. We believe these changes improve clarity and consistency.

Reviewer 2 Report

Comments and Suggestions for Authors

This systematic review and meta-analysis addresses a highly relevant and timely topic in lower limb amputation rehabilitation: the efficacy of Virtual Reality (VR) interventions. The finding of a significant positive effect on postural stability (with ) and the lack of a significant effect on walking capacity (with high heterogeneity, ) provide critical information for clinical practice and future research.

However, the manuscript requires substantial methodological and reporting enhancements. Specific focus must be placed on clarifying potential clerical errors, deeply investigating the sources of heterogeneity, and strengthening the discussion of clinical applicability.

1. Clarity and Reporting (Methodology Section)

  •  

    Critical Date Clarification: The methodology states the search spanned from "July 2024 to June 2025". This is highly unusual and suggests either a major clerical error in the year or an error in the month range, as the manuscript is currently being reviewed in October 2025.

    • Question/Improvement: Please correct this date range immediately (e.g., to July 2010 to June 2024, or the date of submission) and re-verify the full list of included studies. If this is a reference to a prospective registration or an unusual reporting requirement, it must be explicitly justified and clarified to ensure the integrity of the search process

  •  

    Database Scope Justification: The review utilized PubMed, Scopus, and Web of Science.

    • Question/Improvement: Why were other essential biomedical and allied health databases, such as Embase and CINAHL, excluded? Given the heterogeneity and small number of included studies , the exclusion of these major databases could lead to a significant selection bias and should be justified or rectified in the revised manuscript.

  •  

    VR Classification Reporting: The distinction between low and moderate immersion VR is appropriately defined in the methods.

    • Improvement: This critical detail is currently missing from Table 2, which summarizes the intervention characteristics. Please add a column to Table 2 explicitly detailing the "Level of Immersion" (e.g., Low, Moderate, Immersive/High, or Exergame/Wii) for each VR intervention to allow readers to interpret the results based on the type of technology used.

2. Results and Statistical Interpretation

High Heterogeneity in Walking Capacity: The meta-analysis for walking capacity showed no significant overall effect () but reported substantial heterogeneity ().

Improvement: The manuscript correctly states the heterogeneity limits generalizability. The Discussion must be expanded to deeply investigate the potential sources of this heterogeneity in the Walking Capacity subgroup (1.1).

      • Did the included studies use different types of measures (e.g., 6MWT vs. 2MWT) that were inappropriately pooled in the total effect? (It seems the analysis appropriately separates 6MWT and 2MWT in the forest plot (Figure 3) but the for the Total effect is .)

      • Given the high and values, a subgroup analysis based on the Classification of exercise intervention (e.g., 'Mixed Modality' vs. 'Biomechanical' from Table 2) or by Intervention Durationshould be considered in a revised metanalysis to explore the source of varian

  •  

    Clinical Significance of Postural Stability: The meta-analysis for postural stability was significant () with no heterogeneity ().

    Question/Improvement: The pooled mean difference (MD) is (95% CI: ). Given the lower bound of the CI is close to zero, please discuss the Clinical Significance of this finding in the Discussion section. What instruments were pooled for this result (e.g., Berg Balance Scale, TUG, FSST), and what does an unit change represent in terms of a Minimal Clinically Important Difference (MCID)? This is essential for clinical utility

3. Discussion and Limitations

Justification for Lack of Walking Improvement: The discussion hypothesizes that the lack of effect on walking capacity is due to the absence of aerobic activity in VR protocols.

    • Improvement: This is a strong and plausible hypothesis. Please strengthen this argument by referring back to the "Classification of exercise interventions" in Table 2. Point out specifically how many of the protocols classified as "Mixed Modality" or "Biomechanical" (e.g., Wii Fit training or Exergames ) explicitly lacked a structured, progressive aerobic component compared to conventional rehabilitation.

      Risk of Bias (RoB) Interpretation: The authors correctly note the methodological limitations, specifically the lack of blinding among professionals.

      Improvement: In the discussion of RoB 2 (Figure 2), highlight that the inability to blind participants in VR interventions is an inherent limitation of this technology, which often contributes to "Deviations from the intended interventions" (D2) and "Measurement of the outcome" (D4) biases. Emphasize that future trials must prioritize the blinding of outcome assessors (D4) to maintain high methodological quality.

      Cost and Accessibility: The high cost and technical demanding nature of immersive systems are noted as a limitation.

      Improvement: The discussion should include a recommendation for future research to conduct cost-effectiveness analyses comparing low-cost/non-immersive systems (like the Wii Fit used in two studies ) versus high-cost/immersive systems versus conventional therapy. This is crucial for real-world translation and policy-making in healthcare.

Author Response

Dear Editor and Reviewers,

Please find a revision of our manuscript entitled " Effects of Virtual Reality Interventions on Motor Function Rehabilitation in Lower Limb Amputees, A Systematic Review and Metanalysis".

We would like to thank the reviewers for their comments. Changes have been placed highlighted in the revised manuscript. An itemized point-by-point response to comments is presented below.

We have conducted a thorough review of the manuscript’s English language to address and correct the most frequently observed grammatical errors

This systematic review and meta-analysis addresses a highly relevant and timely topic in lower limb amputation rehabilitation: the efficacy of Virtual Reality (VR) interventions. The finding of a significant positive effect on postural stability (with I2=0%) and the lack of a significant effect on walking capacity (with high heterogeneity, I2=75%) provide critical information for clinical practice and future research.

However, the manuscript requires substantial methodological and reporting enhancements. Specific focus must be placed on clarifying potential clerical errors, deeply investigating the sources of heterogeneity, and strengthening the discussion of clinical applicability.

  1. Clarity and Reporting (Methodology Section)

Comment 1. Critical Date Clarification: The methodology states the search spanned from "July 2024 to June 2025". This is highly unusual and suggests either a major clerical error in the year or an error in the month range, as the manuscript is currently being reviewed in October 2025.

Response 1. We thank the reviewer for noting this clerical error. The search period has been corrected to indicate that studies were included from the inception of each database up to July 2025, with the database search conducted between June and July 2025. The revised version now accurately reflects this information in the Methods section.

Comment 2. Question/Improvement: Please correct this date range immediately (e.g., to July 2010 to June 2024, or the date of submission) and re-verify the full list of included studies. If this is a reference to a prospective registration or an unusual reporting requirement, it must be explicitly justified and clarified to ensure the integrity of the search process

Response 2. We appreciate the reviewer’s careful observation. The date range in the Methods section has been corrected to accurately reflect the intended search period. The systematic search was conducted across PubMed, Scopus, and Web of Science, including all records from database inception to July 2025. This correction clarifies that the review includes studies published up to the date of manuscript submission. The full list of included studies has been re-verified to ensure consistency with the updated time frame.

Database Scope Justification: The review utilized PubMed, Scopus, and Web of Science.

Comment 3. Question/Improvement: Why were other essential biomedical and allied health databases, such as Embase and CINAHL, excluded? Given the heterogeneity and small number of included studies (n=6), the exclusion of these major databases could lead to a significant selection bias and should be justified or rectified in the revised manuscript.

Response 3. We appreciate this observation. As noted in the Limitations section, the search was restricted to PubMed, Scopus, and Web of Science, which already provided a large number of records. Many of these studies were subsequently excluded for not meeting the inclusion criteria. Although this approach ensured methodological rigor, we acknowledge that the exclusion of other health science databases (e.g., CINAHL, PEDro, Embase) may represent a potential limitation of our review.

Comment 4. VR Classification Reporting: The distinction between low and moderate immersion VR is appropriately defined in the methods. Improvement: This critical detail is currently missing from Table 2, which summarizes the intervention characteristics. Please add a column to Table 2 explicitly detailing the "Level of Immersion" (e.g., Low, Moderate, Immersive/High, or Exergame/Wii) for each VR intervention to allow readers to interpret the results based on the type of technology used.

Response 4. We thank the reviewer for the valuable suggestion. We have added a new column to Table 2 explicitly detailing the “Level of Immersion” for each VR intervention, classified as Low, Moderate, Immersive/High, or Exergame/Wii, according to the criteria described in the Methods section. This addition allows readers to interpret the results based on the type of VR technology used.

Comment 5. Results and Statistical Interpretation. High Heterogeneity in Walking Capacity: The meta-analysis for walking capacity showed no significant overall effect (P=0.37) but reported substantial heterogeneity (I2=75%). Improvement: The manuscript correctly states the heterogeneity limits generalizability. The Discussion must be expanded to deeply investigate the potential sources of this heterogeneity in the Walking Capacity subgroup (1.1).

Response 5. Both studies included in the meta-analysis subgroup used similar intervention modalities. However, in Sahan’s study, the control group showed greater improvements in walking capacity than the experimental group, likely because the control intervention included aerobic activities that may have enhanced gait function. This point has been addressed in the Discussion section.

Comment 6. Did the included studies use different types of measures (e.g., 6MWT vs. 2MWT) that were inappropriately pooled in the total effect? (It seems the analysis appropriately separates 6MWT and 2MWT in the forest plot (Figure 3) but the I2 for the Total effect is 75%.)

Response 6. Thank you very much for your insightful comment. In the revised manuscript, we have clarified in the Methods section that studies reporting the 6-Minute Walk Test (6MWT) and the 2-Minute Walk Test (2MWT) were analyzed separately in the forest plots.

For the “Total effect,” we used the Standardized Mean Difference (SMD) to combine these measures, as both tests assess the same construct, walking capacity, but use different scales. A Random-Effects (RE) model was also applied due to the substantial heterogeneity observed (I² = 75%).

We have updated the analysis accordingly and provided an updated figure. The high heterogeneity likely reflects differences in test type (2MWT vs. 6MWT), intervention duration, and variability in control group activities across studies. This has been further discussed in the limitation section, where we note that such variability limits the generalizability of the results.

Comment 7. Given the high χ2 and I2 values, a subgroup analysis based on the Classification of exercise intervention (e.g., 'Mixed Modality' vs. 'Biomechanical' from Table 2) or by Intervention Duration should be considered in a revised metanalysis to explore the source of varian

Response 7. Thank you for your comment. We agree that the high χ² and I² values indicate substantial heterogeneity in the walking capacity analysis. However, there is considerable variability among the interventions included in both the experimental and control groups. Therefore, grouping the studies by intervention modality did not reduce heterogeneity; in fact, it increased it, as variability within subgroups remained high. This suggests that the main source of heterogeneity lies in the differences between study protocols rather than the classification of VR interventions. We have included this clarification in the Limitations section.

Comment 8. Clinical Significance of Postural Stability: The meta-analysis for postural stability was significant (P=0.03) with no heterogeneity (I2=0%). Question/Improvement: The pooled mean difference (MD) is 8.03 (95% CI: 0.98,15.07). Given the lower bound of the CI is close to zero, please discuss the Clinical Significance of this finding in the Discussion section. What instruments were pooled for this result (e.g., Berg Balance Scale, TUG, FSST), and what does an 8.03 unit change represent in terms of a Minimal Clinically Important Difference (MCID)? This is essential for clinical utility

Response 8. Thank you for your valuable observation. We have clarified in the Discussion section that all studies included in the postural stability meta-analysis used the Activities-specific Balance Confidence (ABC) Scale as the outcome measure. The pooled mean difference of 8.03 points indicates a statistically significant improvement in balance confidence in favor of the experimental group. However, we also acknowledge that this change is below the minimal clinically important difference (MCID) reported in previous studies, which ranges from 8.64 to 16.94 points depending on the population (Montilla Ibañez et al. 2016). Therefore, while the result demonstrates a meaningful trend toward improvement, its clinical significance should be interpreted with caution.  

Comment 9. Discussion and Limitation. Justification for Lack of Walking Improvement: The discussion hypothesizes that the lack of effect on walking capacity is due to the absence of aerobic activity in VR protocols. Improvement: This is a strong and plausible hypothesis. Please strengthen this argument by referring back to the "Classification of exercise interventions" in Table 2. Point out specifically how many of the protocols classified as "Mixed Modality" or "Biomechanical" (e.g., Wii Fit training or Exergames ) explicitly lacked a structured, progressive aerobic component compared to conventional rehabilitation.

Response 9. Thank you for your valuable comment. We have strengthened the discussion section as suggested. Specifically, we referred back to the Classification of exercise interventions presented in Table 2 and clarified that most of the protocols categorized as Mixed Modality or Biomechanical, only the control group of Sahan et al study. This information has been added to the Discussion to better justify the lack of improvement in walking capacity compared with conventional rehabilitation programs.

Comment 10. Risk of Bias (RoB) Interpretation: The authors correctly note the methodological limitations, specifically the lack of blinding among professionals. Improvement: In the discussion of RoB 2 (Figure 2), highlight that the inability to blind participants in VR interventions is an inherent limitation of this technology, which often contributes to "Deviations from the intended interventions" (D2) and "Measurement of the outcome" (D4) biases. Emphasize that future trials must prioritize the blinding of outcome assessors (D4) to maintain high methodological quality.

Response 10. We thank the reviewer for the insightful comments. We have updated the discussion and limitation section to address the points raised. Specifically, we have highlighted that the inability to blind participants in VR interventions represents an inherent limitation of this technology. Furthermore, we have emphasized that future trials should prioritize the blinding of outcome assessors to ensure high methodological quality.

Comment 10. Cost and Accessibility: The high cost and technical demanding nature of immersive systems are noted as a limitation. Improvement: The discussion should include a recommendation for future research to conduct cost-effectiveness analyses comparing low-cost/non-immersive systems (like the Wii Fit used in two studies ) versus high-cost/immersive systems versus conventional therapy. This is crucial for real-world translation and policy-making in healthcare.

Response 10. We thank the reviewer for this insightful comment regarding cost and accessibility. We have expanded the Discussion section to highlight that the majority of the studies included in our review employed lower-cost virtual reality systems. We also acknowledge the diversity of opinions in the literature regarding costs, noting that some studies report potential reductions in professional workload and transportation costs when interventions are delivered remotely over distance. Furthermore, we have added a recommendation for future research to conduct cost-effectiveness analyses comparing low-cost/non-immersive systems, high-cost immersive systems, and conventional therapy, emphasizing the importance of such analyses for real-world implementation and healthcare policy-making.

Round 2

Reviewer 2 Report

Comments and Suggestions for Authors

The authors have thoroughly revised and improved the manuscript. This has considerably increased its methodological quality and scientific rigor. Excellent work.